# Outcomes of Lower Extremity Endovascular Revascularization: Potential Predictors and Prevention Strategies

**DOI:** 10.3390/ijms22042002

**Published:** 2021-02-18

**Authors:** Federico Biscetti, Elisabetta Nardella, Maria Margherita Rando, Andrea Leonardo Cecchini, Antonio Gasbarrini, Massimo Massetti, Andrea Flex

**Affiliations:** 1Fondazione Policlinico Universitario A. Gemelli IRCCS, 00168 Roma, Italy; m.margheritarando@gmail.com (M.M.R.); antonio.gasbarrini@unicatt.it (A.G.); massimo.massetti@unicatt.it (M.M.); andrea.flex@unicatt.it (A.F.); 2Cardiovascular Internal Medicine Unit, Fondazione Policlinico Universitario A. Gemelli IRCCS, 00168 Roma, Italy; 3Department of Medical and Surgical Sciences, Università Cattolica del Sacro Cuore, 00168 Roma, Italy; elisabetta.nardella@gmail.com (E.N.); alcech92@gmail.com (A.L.C.); 4Department of Cardiovascular Sciences, Fondazione Policlinico Universitario A. Gemelli IRCCS, 00168 Roma, Italy

**Keywords:** peripheral artery disease (PAD), chronic limb-threatening ischemia (CLTI), revascularization

## Abstract

Peripheral artery disease (PAD) is a manifestation of atherosclerosis, which may affect arteries of the lower extremities. The most dangerous PAD complication is chronic limb-threatening ischemia (CLTI). Without revascularization, CLTI often causes limb loss. However, neither open surgical revascularization nor endovascular treatment (EVT) ensure long-term success and freedom from restenosis and revascularization failure. In recent years, EVT has gained growing acceptance among all vascular specialties, becoming the primary approach of revascularization in patients with CLTI. In clinical practice, different clinical outcomes after EVT in patients with similar comorbidities undergoing the same procedure (in terms of revascularization technique and localization of the disease) cause unsolved issues that need to be addressed. Nowadays, risk management of revascularization failure is one of the major challenges in the vascular field. The aim of this literature review is to identify potential predictors for lower extremity endovascular revascularization outcomes and possible prevention strategies.

## 1. Introduction

Peripheral artery disease (PAD) is a manifestation of atherosclerosis disease, which may affect arteries in the lower extremities. It is the third most common manifestation of atherosclerosis after coronary artery disease (CAD) and stroke [1]. Fatigue, atypical leg pain, and cramping during ambulation—known as intermittent claudication (IC)—are typical signs of symptomatic PAD [2]. The most dangerous complication of PAD is critical limb ischemia (CLI), which is nowadays defined as chronic limb-threatening ischemia (CLTI) to include a wider and more heterogeneous group of patients with varying degrees of ischemia. The diagnosis of CLTI requires a documented atherosclerotic PAD associated with ischemic rest pain or tissue loss (ulceration or gangrene) [3].

It is estimated that over 230 million persons have PAD worldwide [4]. PAD may implicate a reduced functional capacity with an inevitable worsening in quality of life. Moreover, it is also associated with an increased risk for cardiovascular morbidity and mortality [5]. Therefore, PAD represents a major public health problem with a significant impact on healthcare causing a notable economic burden [6].

Without revascularization, CLTI often results in limb loss. However, neither open surgical revascularization nor endovascular treatment (EVT) guarantee treatment success and freedom from restenosis and revascularization failure [7,8]. In recent years, EVT has gained growing acceptance among all vascular specialties, becoming the primary approach of revascularization in patients with CLTI [9,10,11]. In clinical practice, different clinical outcomes after EVT in patients with similar comorbidities undergoing the same procedure (in terms of revascularization technique and localization of the disease) cause unsolved issues that need to be addressed [12]. Nowadays, the risk management of revascularization failure is one of the major challenges in the vascular field. The aim of this literature review is to identify potential predictors of lower extremity endovascular revascularization (LER) failure and possible prevention strategies.

## 2. Definitions of Revascularization Failure

Effective revascularization is defined by criteria for both anatomic and hemodynamic success. Indeed, a procedure may be technically successful and clinically unsuccessful. Some approaches may produce a better anatomic result without improving hemodynamics [13].

The endpoints of revascularization are classified into three categories: primary patency, primary-assisted patency, and secondary patency [14]. Primary patency is defined as continuous patency without directly performing an intervention or procedure. Primary-assisted patency is defined as uninterrupted patency, but it is maintained by prophylactic intervention. Secondary patency is the time from the procedure to restenosis [7]. The definition of restenosis is when the luminal diameter is narrowed by over 50% or the cross-sectional area is reduced by over 75% [15]. The complex pathophysiologic mechanism remains incompletely understood. Angioplasty, stent placement, or atherectomy represent a mechanical injury to the vessel, which locally reacts with an inflammatory response. This process may lead to restenosis because of intimal thickening and an increased extracellular matrix [16,17]. Neointimal hyperplasia acts as a main restenosis trigger after stenting. Instead, restenosis after angioplasty or atherectomy results from a combination of constrictive arterial remodeling along with neointimal hyperplasia [18,19,20,21,22].

Currently, endovascular angioplasty includes different techniques such as plain old balloon angioplasty, cryoplasty, cutting balloon angioplasty, and medicated balloon angioplasty. Self-expanding and balloon-expandable are two general types of stents. They are available with or without polytetrafluoroethylene cover, are bioabsorbable, and drug eluting [15]. These technologies were developed mainly to reduce restenosis rates after EVT. However, the discussion of technical considerations is in intentional scientific articles limited and therefore, also in this article restrictedly addressed [3].

Other important outcomes after LER are major adverse limb events (MALE)—defined as composite of acute limb ischemia, major vascular amputations, limb-threatening ischemia leading to urgent revascularization—and major adverse cardiovascular events (MACE)—defined as a composite of acute myocardial infarction, stroke, transient ischemic attack, and cardiovascular death [23,24].

## 3. Revascularization: When Is It Appropriate?

The goals of EVT in patients affected by PAD are pain relief, wound healing, and functional limb preservation. Nevertheless, revascularization may cause morbidity, which is correlated to many hospital admissions, continuous outpatient care, and significant treatment and health care costs, as well as mortality. Some patients can be appropriately treated with primary amputation or palliative care after multidisciplinary decision-making [3]. Therefore, patients for whom EVT may be beneficial need to be identified to avoid potential failure. Predicting functional outcomes after revascularization is difficult, particularly in patients who are severely deconditioned. The Global Vascular Guidelines (GVG) suggests a structured approach based on a three-step process: patient risk estimation, limb severity, and anatomic pattern of disease (PLAN) [3].

### 3.1. Patient Risk Estimation

CLTI affects patients with advanced age and multiple comorbidities. In this setting, estimation of operative risk and life expectancy is pivotal. Preoperative cardiac and anesthetic evaluation before limb revascularization is mandatory [3,25,26]. Several procedural risk factors have been identified for the CLTI population. They include advanced age (over 75 or 80 years), CAD, congestive heart failure, diabetes mellitus (DM), chronic kidney disease (CKD), smoking, cerebrovascular disease, tissue loss, body max index (BMI), dementia, functional status, and frailty. In recent years, multiple risk stratification tools have been retrospectively developed for patients who underwent surgical revascularization [3,27,28,29,30,31,32,33,34], but none has been tested prospectively and endorsed by international guidelines.

### 3.2. Limb Severity

Patients affected by CLTI present a wide-ranging disease severity. The GVG recommends the Society for Vascular Surgery (SVS) threatened limb classification system integrating wound severity, ischemia, and foot infection (WIfI) to stage CLTI [35,36,37,38]. The classification grades each component from 0 to 3, and a higher number indicates an increasing severity. Sixty-four theoretical patient combinations exist, with four possible clinical stages, ranging from a very low to a high amputation risk and revascularization benefit, respectively. WIfI has shown accuracy in predicting amputation risk and revascularization benefit [39,40]. Mayor et al. demonstrated that WIfI allows identifying which group of patients affected by CLTI may benefit from revascularization [41].

### 3.3. Anatomic Pattern of Disease

The third step of PLAN is the definition of an anatomic disease pattern. The traditional anatomic classification systems for PAD are lesion or segment focused [42,43]. The GVG propose a new approach named global limb anatomic staging system (GLASS) to integrate patterns of disease, hemodynamic improvement after treatment, anatomic durability, clinical stage, and outcomes [3]. This limb anatomic system introduces two novel concepts, which are the target arterial path (TAP) and the estimated limb-based patency (LBP), which is defined as patency along the TAP. GLASS describes three complexity stages for intervention derived from combined femoropopliteal and infrapopliteal GLASS grades (1–4), which represent increasing severity and disease complexity along the TAP [3]. GLASS stages are related to disease complexity for endovascular treatment (EVT), to immediate technical failure, and to one year LBP of the TAP [44]. Kodama et al. examined the relationship between GLASS and clinical outcomes in the Bypass versus Angioplasty in Severe Ischaemia of the Leg (BASIL)-1 trial patient cohort and found that GLASS was associated with EVT outcomes but not with bypass surgery [44]. This new limb-integrated system may facilitate both shared decision-making and CLTI patient stratification.

## 4. The Unsolved Conundrum of Controlling Risk Factors

Despite GVG providing an innovative approach with PLAN, the incidence of LER failure is still high.

The PAD population is characterized by some risk factors, which contribute to develop CLTI and cause worse outcomes after LER. For some of these risk factors, established therapeutic and pharmacological patient management strategies exist. For example, international guidelines recommend the use of moderate- to high-intensity statin therapy, because it is associated with significantly improved survival and with a lower MALE rate in patients undergoing revascularization for CLTI [3,45]. The adverse impact of tobacco use on cardiovascular outcomes has been well established. It is universally recognized that stopping smoking and adopting a healthy diet, weight control, and regular exercise are very important for both life and limb [3]. Nevertheless, current therapeutic strategies, in some conditions, are not associated with reduced LER failure. These conditions are well-recognized, but their pathogenic mechanism is not fully understood in patients with CLTI undergoing EVT. Therefore, it is essential to improve knowledge and to identify effective strategies.

### 4.1. Management of Hypertension

Although optimal blood pressure control for patients with CLTI has not been established, it is unanimously accepted that hypertension control reduces MACE in patients with PAD [3,46]. Moreover, several studies documented that angiotensin-converting enzyme inhibitors (ACEI) are associated with reduced MACE [47,48,49]. A study showed that patients with CLTI treated with ACEI and LER resulted in statistically significant more amputations, which remained significant after adjustment at 1 year [50]. Other studies found that ACEI use was associated with an increased rate of re-intervention [51]. Recently, Khan et al. [52] showed that ACEI use is associated with improved limb salvage rates in CLTI patients undergoing infrapopliteal but not femoropopliteal interventions, but no effects of ACEI on patency rates were found. The use of beta-blockers is traditionally contraindicated in patients with PAD due to concerns regarding reduced peripheral perfusion [53,54]. A Cochrane and a meta-analysis showed that beta-blockers do not worsen leg ischemia in patients with IC [55,56]. Long-term results of large prospective trials in patients with CLTI treated with ACEI and beta-blockers treatment are required to better understand the effects of these drugs on this population.

### 4.2. Glycemic Control 

Diabetes mellitus (DM) represents the major risk factors of PAD [57,58,59]. A meta-analysis highlights that DM, independently from other major risk factors, increases vascular risk in both men and women [60]. Diabetes leads to endothelial dysfunction, vascular smooth muscle cell proliferation, and platelet activation [61]. Diabetic patients may have aggressive below-the-knee PAD and chronic CLTI [3,62,63]. Patients with PAD and DM have a major amputation risk which is 10 times higher than those without DM [64,65]. The extent of vascular disease is related to the duration and severity of hyperglycemia [66,67,68]. Current clinical guidelines recommended adequate glycemic control in diabetic patients to manage PAD [3,69,70]. However, the effects of intensive glucose control on the macrovascular outcomes among diabetic patients are controversial [71]. In randomized trials, tight glucose control in diabetic patients did not reduce MACE compared with standard glucose management [72,73,74]. Nevertheless, several post hoc analyses demonstrated that higher glycated hemoglobin (HbA1c) levels were associated with increased MACE [75] and MALE risks [76,77]. Two single-center studies suggested that poor glycemic control at the time of peripheral angioplasty was associated with worse clinical outcomes in patients with CLTI [78,79]. A retrospective analysis of US veterans affected by CLTI undergoing EVT showed that patients with poor glycemic control were at higher risk of amputation and MALE than those with good glycemic control [66]. Recently, Cha et al. reported, using data of a Korean multicenter retrospective registry cohort, the association between the preprocedural glycemic control based on HbA1c during index admission and clinical outcomes in diabetic patients undergoing EVT for PAD [80]. The suboptimal glycemic control group (HbA1c ≥ 7.0) had a higher MALE incidence compared with the optimal glycemic control group (HbA1c < 7.0). Suboptimal glycemic control was an independent predictor of re-intervention during the follow-up period after EVT in diabetic patients with PAD [80]. The authors further analyzed data dividing into the presence of IC and CLTI. They found that suboptimal pre-procedural HbA1c did not affect the outcome in CLTI patients, whereas the incidence of MALE increased significantly among those with IC. This result is in accordance with our experience in which we found no correlation between HbA1c and fasting blood glucose (FBG) with the outcomes of diabetic patients with CLTI undergoing EVT [12]. Singh et al. hypothesized that disease severity is particularly critical at the time of EVT in which a better glycemic index has no discernable effect on clinical outcomes [81]. Moreover, several pieces of evidence suggested that glycemic variability may play a very important role in the atherosclerosis pathogenesis, independent of HbA1c level [82,83,84].

Large prospective randomized trials will be needed to better understand optimal preoperative glycemic control in diabetic patients with PAD undergoing EVT.

Moreover, evidence showed that frequent HbA1c monitoring, in the outpatient setting, improved long-term limb salvage and readmission rates in patients undergoing EVT [85]. Currently, the optimal target of HbA1c or FBG for this population is unknown, and the hypoglycemic pharmacological strategies are not standardized. Recent studies have shown that novel glucose-lowering agents may reduce the risk of cardiovascular events in patients with atherosclerotic disease [86,87]. However, increased risk of lower limb amputation has been described for sodium-glucose cotransporter type-2 inhibitors in patients with PAD [88,89]. The mechanism is unknown. Potier et al. described a possible correlation with diuretics and a significant risk increase in lower limb events, especially lower limb amputation [90]. Several studies examined the effects of dipeptidyl peptidase 4 inhibitors use on the risk of MACE (for a review, see [91]), but the impact on patients with CLTI and MALE has not been evaluated. Indeed, studies on hypoglycemic drugs classes are designed to evaluate the effects on outcomes of MACE. Further investigations are required, which evaluate MALE outcomes.

### 4.3. Chronic Kidney Disease 

CKD is common in patients affected by PAD [92,93]. PAD and CKD have the same risk factors, such as diabetes, hypertension, dyslipidemia, advanced age, and smoking [94]. At the same time, PAD is prevalent in patients with CKD [95], especially among those with end-stage renal disease (ESRD) [96,97]. The 2005 kidney disease outcomes quality initiative guidelines recommend screening for PAD at the time of dialysis initiation [98,99].

Although CKD is an independent risk factor for PAD [98,100], it remains underestimated in current clinic guidelines [101]. Moreover, renal impairment is an important predictor of poor outcomes [94]. In a recent analysis, Smilowitz et al. examined a large representative national cohort from the United States undergoing surgical or endovascular revascularization for PAD with and without CKD or ESRD [102]. They found that CKD and ESRD were associated with a 74% increased risk of perioperative MACE in patients undergoing both endovascular and surgical lower extremity revascularization. Lower extremity amputation occurred more frequently in patients with PAD and renal disease. Moreover, patients with renal disease required more hospital readmission within 6 months compared to those without renal disease [102]. In this analysis, the most common indications for hospital readmission were cardiovascular disease, endocrine/metabolic issues, and infectious complications. Indeed, CKD is a condition related to increased infection risk owing to functional immunosuppression ascribed to decreased glomerular filtration rate (GFR) and albuminuria [103].

A cross-sectional analysis of the US Renal Data System identified kidney-specific risk factors of PAD such as dialysis duration, hypoalbuminemia, hyperphosphatemia, inflammation, and malnutrition [104]. The typical feature of PAD in patients with CKD is diffuse calcified atherosclerosis caused by inflammation with oxidative stress, impaired angiogenesis, and uremic vasculopathy [105]. The presence of calcified lesions, in particular arterial intima calcification, represents a challenge for both surgical and endovascular therapy [106]. Therefore, calcification is a predictor for restenosis and revascularization failure [94]. EVT is often ineffective because of dissection and perforation risks, as well as the necessity to perform adjuvant atherectomy that may lead to distal embolization. No large randomized studies compared percutaneous versus surgical revascularization techniques in CKD patients with PAD. However, most of the available studies suggest that EVT is a reasonable first-line treatment in patients with CLTI and CKD, although outcomes are worse among patients with higher rates of repeated percutaneous angioplasty, subsequent surgical revascularization, or limb loss and death [107,108,109,110,111,112,113,114,115]. Perioperative morbidity and mortality are higher among CKD and dialyzed patients undergoing surgical procedures [95]; therefore, percutaneous methods are preferred [30,116,117,118].

Prevention, early diagnosis, and treatment of risk factors are key elements to improve outcomes in patients with CLTI and renal impairment, as well as the development of novel endovascular technologies to address vascular calcification.

### 4.4. Functional Status 

Currently, no validated tools exist to assess grade and consider functional status, which is a major consideration during limb salvage decision. Khan et al. propose a functional outpatient score in patients affected by CLTI using a numeric scoring system in which score 0 indicates that the patients can walk outside his/her home with or without assistance; score 1 describes a patient who is able to walk only at home with/without assistance, score 2 indicates a patient with minimal ambulation ability, limbs are used for transfers, and score 3 describes a bed-bound patient [119].

The authors recommend applying this functional outpatient status as an adjunctive clinical decision-making tool to the current WIfI system to determine who may benefit most from revascularization with the goal of preserving baseline outpatient status [119]. Indeed, WIfI stages the limb, but not the patient, which is analogous to TNM staging for cancer. The goal is to not only to save limbs but also to guarantee optimal functional outcome. How to solve the functional status to unravel the CLTI puzzle is intricate and requires further studies with data on long-term outcomes [120].

## 5. The Inflammatory State in CLTI

The progression of atherosclerosis is characterized by an inflammatory reaction orchestrated by several molecules belonging to different families of inflammatory mediators, such as cytokines, chemokines, adhesion molecules, and proteolytic enzymes [121,122,123]. Similarly, diabetes and its complications cause a chronic, low-grade inflammatory status [124,125]. Based on this knowledge, Signorelli et al. investigated the plasma levels of inflammatory markers such as interleukin (IL)-6 and tumor necrosis factor (TNF)-α in patients with PAD and in healthy controls [126]. They found, in agreement with previous findings [127], that patients with PAD show an inflammation marker profile different from that of controls. In literature, data regarding markers of inflammation and EVT outcomes in patients with PAD are presented. Barani et al. found that the inflammatory mediators IL-6, TNF-α, N, and high-sensitivity C-reactive protein (CRP) were associated with 1-year mortality in CLTI patients [128]. For TNF-α and N, this association was independent of other variables such as age, sex, gangrene, active treatment, lipid-lowering therapy, leukocyte count, renal function, and cholesterol levels. Schillinger et al. demonstrated that pre- and post-intervention CRP levels were associated with restenosis after percutaneous transluminal angioplasty (PTA) of the distal popliteal and tibio-peroneal arteries, which indicates that inflammation plays a crucial role in the pathophysiology process [129]. Higher CRP levels were also found in diabetic patients who underwent PTA of the lower limb [130]. Bleda et al. analyzed the possible association between CRP and fibrinogen before EVT and during 1-year follow-up as well as its variation during the study period [131]. They found a significant correlation between basal levels of CRP and fibrinogen and the incidence of re-intervention, cardiovascular events, and death during follow-up. Moreover, Stone et al. found that after lower extremity endovascular interventions, elevated preprocedural CRP levels are associated with MALE and elevated levels of CRP and BNP are associated with late cardiovascular events [132].

Given these previous data, we hypothesized a correlation between osteoprotegerin (OPG), TNF-α, IL-6, and CRP levels at baseline before EVT and outcomes in patients with diabetes, PAD, and CLTI [12]. Indeed, OPG is a recognized marker of atherosclerosis in diabetic patients, has a role in calcium metabolism, and directly interacts with the vascular endothelium [133,134,135,136]. TNF-α has a pivotal role in the pathway of diabetic atherosclerosis [61]. IL-6 has positive effects on endothelial function and aortic stiffness, as demonstrated by data from patients treated with IL-6 inhibitors [137]. CRP improves foam cell formation in atherosclerotic plaque and promotes platelet adhesion [138]. In our study, we found a significant linear trend between increasing levels of each cytokine and risk of MALE and MACE in patients with diabetes, PAD, and CLTI who underwent an endovascular procedure [12].

In addition, high mobility group box-1 (HMGB-1) is a nuclear protein that controls gene expression and starts pro-inflammatory responses in damaged and necrotic endothelial cells, taking part in inflammatory pathways [139,140,141]. Studies described a relationship between HMGB-1 levels, diabetes, and its complications [142]. Oozawa et al. found increased HMGB1 plasma levels in diabetic patients with PAD [143]. We demonstrated in a large population of diabetic patients that HMGB-1 plasma levels are significantly increased in patients affected by PAD with a positive correlation with clinical severity of vascular damage [136].

Assessment of the inflammatory state, quantifying serum cytokines, may support physicians to identify a subset of patients more susceptible to failure after the EVT.

The possibility that anti-inflammatory therapy can improve outcomes of LER should be investigated.

## 6. Novel Biomarkers in CLTI

In recent years, a growing number of plasma biomarkers have been tested as prognostic tools for diabetic complications [144].

Adipokines are bioactive substances produced by visceral adipose tissue. They balance pro-inflammatory and anti-inflammatory effects of adipose tissue [145]. DM contributes to adipose tissue dysfunction leading to insulin resistance, vascular injury, and consequent vascular disease. Several studies are exploring the possible role of adipokines as biomarkers for diabetic vascular complications. Our group found that omentin-1 serum levels were significantly lower in diabetic patients with PAD than in diabetic patients without PAD [146]. We also showed that the baseline omentin-1 levels were associated with MACE and MALE after LER in diabetic patients with PAD and CLTI [147]. Omentin-1 may help stratify patients to facilitate a more appropriate diagnostic and prognostic assessment.

Sortilin is a protein mainly expressed in hepatocytes implicated in LDL uptake by hepatocytes via an independent LDL-receptor mechanism and by macrophages in the process of foam cell formation in atheroma [148]. Moreover, sortilin plays a role in intracellular cytokine traffic and in platelet activation [149]. Several researchers have examined a possible association among sortilin, atherosclerosis, diabetes, and cardiovascular complications [150]. We observed that sortilin is independently associated in a diabetic population with PAD [151]. We recently showed that sortilin levels were related with MACE and MALE incidence after LER in a diabetic population with PAD and CLTI [152].

Further investigations on a larger cohort are required to confirm our results on the role of omentin-1 and sortilin. Currently, no biomarkers are available in clinical practice to predict LER failure in patients with CLTI. Biomarkers could allow risk stratification and an effective follow-up of these patients.

## 7. The Rationale of Surveillance and Post-Procedural Care

There is still no agreement on follow-up after EVT of CLTI at this time. The rationale of surveillance is to detect and treat revascularization failure before patency loss in order to implement preventive strategies for disease progression while avoiding other cardiovascular events [153]. Indeed, endpoints such as primary-assisted patency will depend on how these lesions are diagnosed and the specialist’s persistency to treat them [7]. Nevertheless, high-quality evidence supporting systematic surveillance are still lacking [153]. The SVS suggests clinical examination, ankle brachial index (ABI), and duplex ultrasound imaging (DUS) within the first month following femoropopliteal artery EVT to provide a post-treatment baseline and evaluate for residual stenosis and continued surveillance at 3 months and then every 6 months [15,154,155,156].

GVG recommend that patients undergoing EVT for CLTI should maintain medical treatments to decelerate atherosclerosis progression and moderate the impact of risk factors [3]. Long-term antiplatelet therapy is pivotal to reduce atherothrombotic events and to improve patency and limb salvage rates after EVT [157,158]. Despite an absence of ascertained evidence, dual antiplatelet therapy (DAPT), aspirin plus clopidogrel, is often prescribed for 1 to 6 months after EVT [159,160]. Further clinical trials are needed to better understand the risks and benefits of DAPT after EVT and to define optimal dosage and treatment duration. No clear recommendation regarding the potential benefit of cilostazol after EVT for CLTI exist [3]. Clearly, lifestyle modifications should be optimized.

## 8. Conclusions

Patients affected by CLTI should be managed by a multidisciplinary team and with a comprehensive approach (Figure 1). Further clinical trials are required to validate and refine PLAN to assess patients’ preoperative risk, limb severity, and anatomic disease patterns. In this review, we identified known factors or conditions for which effective therapeutic strategies do not exist yet. Increased awareness of these predictive conditions of restenosis may guide efforts to improve the assessment of patients with CLTI. All-encompassing tools are required to simplify therapeutic choices in the setting of a complex disease that always arises in patients with several comorbidities.

## Figures and Tables

**Figure 1 ijms-22-02002-f001:**
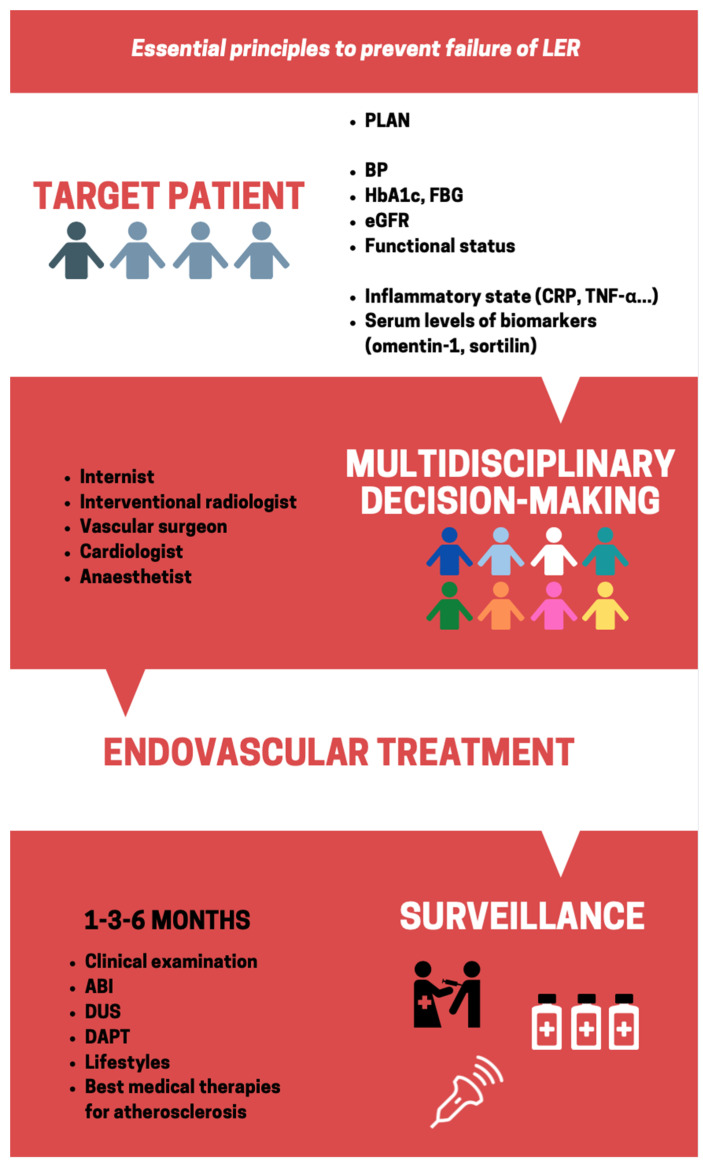
Essential principles to prevent failure of Lower extremity endovascular revascularization (LER). PLAN: Patient risk estimation, Limb severity, ANatomic disease patter; BP: blood pressure; HbA1c: glycated hemoglobin; FBG: fasting blood glucose; eGFR: estimated glomerular filtration rate; CRP: C-reactive protein; TNF-α: tumor necrosis factor-α; ABI: ankle brachial index; DUS: duplex ultra sound; DAPT: dual antiplatelet therapy.

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
