# Peer review of "Outcomes of Lower Extremity Endovascular Revascularization: Potential Predictors and Prevention Strategies"

_ijms, 2021, doi:10.3390/ijms22042002_

Round 1
Reviewer 1 Report
In this nice paper the authors explore the current knowledge on the revascularization procedure and the ability to predict post procedural clinical outcomes in patients affected by peripheral artery disease (PAD) and Chronic limb-threatening ischemia (CTLI).
CLTI is an advanced stage of PAD that has a high risk of developing into a potentially life-threatening condition and resulting in severe disability. Moreover, the prevalence of this disease and the incidence of its complications significantly affect the patients’ quality of life as well as having a significant impact on healthcare costs. The ability to predict the clinical outcome after a revascularization procedure in patients with similar comorbidities and undergoing the same procedure (in terms of technique performed and localization of the disease) is still an unmet aim.
This article conducts a thorough review of current knowledge on the revascularization procedure and the ability to predict post procedural clinical outcomes. The authors provided a detailed list of the main risk predictors of PAD complications and also showed the recent results on new biomarkers for the assessment of MALE and MACE in the PAD population.
Furthermore, the authors showed another limitation in the post-procedural management of these patients, as there is currently no high-quality evidence to support a specific post-treatment surveillance plan.
This review has effectively proposed some factors and conditions that deserve greater awareness for a global evaluation of the patient affected by CTLI and for simplifying the therapeutic approach of such a complex disease in equally complicated patients.
Author Response
We thank the Reviewer for his/her encouraging comments.
Reviewer 2 Report
This review on an interesting topic is grammatically not very well written and therefore not very well readable. I would suggest the authors ask help of a native speaker to rewrite.
In my opinion the title of this review does not entirely cover the content of the manuscript. In particular the sections Functional status, CKD, hypertension, etc in CLTI patients; these section give information about controlling risk factors and the evidence in this perspective. However, only limited information is supplied on (failure of) endovascular therapy in these patients.
The most important concern I have with the current manuscript is its distinctiveness when compared to the clinical practice guideline document: 'Global Vascular Guidelines on the Management of ChronicLimb-Threatening Ischemia' by Conte et al published in 2019.
This guideline descrbes the anatomical scoring systems, the PLAN strategy, controlling risk factors and the evidence on this matter. What will be the added value of this currently proposed manuscript compared to the guideline published in 2019?
The aim of this review is to identify the potential predictors of restenosis risk. I think this aim is not clearly worked out in the different section of this review.
Minor points
Introduction (lines 44 - 46): please rephrase sentence, 'are freedom from...' --> not correct.
Introduction (lines 47-49): please rephrase sentence gramatically
More sentences to follow are gramaticcaly not correct. Please ask native writer to help revise if feasible.
Lines 313 – 317: please rephrase, in my opinion this conclusion cannot be drawn following the (limited) available evidence.
Author Response
Reviewer 2
This review on an interesting topic is grammatically not very well written and therefore not very well readable. I would suggest the authors ask help of a native speaker to rewrite.
Authors’ response
We thank the Reviewer for her/his comment. We would like to take this opportunity to express our sincere thanks to the Reviewer for identifying areas of our manuscript needing modification.
The manuscript has been completely revised by an English native speaker.
Reviewer 2
In my opinion the title of this review does not entirely cover the content of the manuscript. In particular the sections Functional status, CKD, hypertension, etc in CLTI patients; these section give information about controlling risk factors and the evidence in this perspective. However, only limited information is supplied on (failure of) endovascular therapy in these patients.
Authors’ response
We thank the Reviewer for her/his comments. The title has been changed to better reflect the content of the review.
Reviewer 2
The most important concern I have with the current manuscript is its distinctiveness when compared to the clinical practice guideline document: 'Global Vascular Guidelines on the Management of ChronicLimb-Threatening Ischemia' by Conte et al published in 2019.
This guideline descrbes the anatomical scoring systems, the PLAN strategy, controlling risk factors and the evidence on this matter. What will be the added value of this currently proposed manuscript compared to the guideline published in 2019?
The aim of this review is to identify the potential predictors of restenosis risk. I think this aim is not clearly worked out in the different section of this review.
Authors’ response
We thank the Reviewer for her/his comments. In the revised manuscript, we made an effort to bring out the distinctiveness of our manuscript. We proposed some factors and conditions that deserve greater awareness for a global evaluation of patients affected by CLTI and simplifying the therapeutic approach of such a complex disease that arises in patients with several comorbidities.
Reviewer 2
Minor points
Introduction (lines 44 - 46): please rephrase sentence, 'are freedom from...' --> not correct.
Introduction (lines 47-49): please rephrase sentence grammatically.
More sentences to follow are gramaticcaly not correct. Please ask native writer to help revise if feasible.
Authors’ response
We thank the Reviewer for her/his comments. The sentences at lines 44-46 and 47-49 have been rewritten.
The manuscript has been revised by an English native speaker.
Reviewer 2
Lines 313 – 317: please rephrase, in my opinion this conclusion cannot be drawn following the (limited) available evidence.
Authors’ response
We thank the Reviewer for her/his comment. The sentence has been rewritten underlining hypothetical sense.
Round 2
Reviewer 2 Report
"An effort was made to bring out the distinctiveness of this manuscript". However, sentences added or the way this effort was made were not supplied. When reading the revised version of the manuscript, in my opinion, the title, textual/grammatical errors and sections that have high similarity with other papers were adressed. However, I'm still worried this manuscript does not add enough to the current literature as the guideline published in 2019 covers most of its content.
Author Response
We thank the Reviewer for her/his comment.
Global Vascular Guidelines (GVG) on the Management of Chronic Limb-Threatening Ischemia (Conte et al, 2019) are focused on definition, evaluation and management of CLTI.
Our manuscript is focused on the issues about the clinical outcomes after a revascularization procedure in patients with CLTI. In our clinical practice, hospital readmissions of patients who underwent endovascular treatment (EVT) for CLTI for restenosis or, even worse, for MALE or MACE are common. In order to distinguish the focus of our manuscript from GVG, we concentrated on the importance of predicting and possibly preventing such specific clinical events both in the title and in the text (e.g. In clinical practice, different clinical outcomes after EVT in patients with similar comorbidities undergoing the same procedure (in terms of revascularization technique and localization of the disease) cause an unsolved issue, which needs to be addressed […]).
GVG did not focus on discussing this unmet clinical need. Hence, our aim is to raise the issue of the importance of these detrimental complications.
GVG widely discussed about risk factors for PAD and medical management of patients with CLTI. Among these risk factors, we identified some well-recognized conditions for which current therapeutic strategies are not associated with reduced LER failure. ([…] Despite GVG provide an innovative approach with PLAN, the incidence of LER failure is still high [...] Nevertheless, current therapeutic strategies, in some conditions, are not associated with reduced LER failure. These conditions are well known, but their pathogenic mechanism is not fully understood in patients with CLTI undergoing EVT. Therefore, it is essential to improve knowledge and to identify effective strategies […]).
In our manuscript, Management of hypertension (paragraph 4.1), Glycemic control (paragraph 4.2), Chronic kidney disease (paragraph 4.3), Functional status (paragraph 4.4) are not discussed as risk factors of PAD or CLTI, but as potential predictors of LER outcomes. In order to distinguish the focus of our manuscript from GVG, we changed the title of the paragraphs.
Finally, we showed recent results on the role of inflammatory state in CLTI (paragraph 5) and new biomarkers for diagnostic and prognostic assessment of patients with CLTI (paragraph 6). These topics were not debated by GVG.
We thank the Reviewer for allowing a substantial improvement of the manuscript.
Round 3
Reviewer 2 Report
The last response of the authors regarding this manuscript is more clear in what exactly is the added value of the manuxcript. On the whole the quality and organization of the manuscript is well improved.